# The Development of a Novel Nitrate Portable Measurement System Based on a UV Paired Diode–Photodiode

**DOI:** 10.3390/s24165367

**Published:** 2024-08-20

**Authors:** Samuel Fernandes, Mouhaydine Tlemçani, Daniele Bortoli, Manuel Feliciano, Maria Elmina Lopes

**Affiliations:** 1Department of Mechatronics Engineering, School of Science and Technology, Universidade de Évora, 7000-671 Évora, Portugal; tlem@uevora.pt; 2Instrumentation and Control Laboratory (ICL), Insititute of Earth Sciences (ICT), Universidade de Évora, 7000-671 Évora, Portugal; db@uevora.pt; 3Physics Department, School of Science and Technology (ECT), Universidade de Évora, 7000-671 Évora, Portugal; 4Earth Remote Sensing Laboratory (EaRSLab), Institute of Earth Sciences (ICT), Universidade de Évora, 7000-671 Évora, Portugal; 5Centro de Investigação de Montanha (CIMO), Instituto Politécnico de Bragança, Campus de Santa Apolónia, 5300-253 Bragança, Portugal; 6Laboratório Associado para a Sustentabilidade e Tecnologia em Regiões de Montanha (SusTEC), Instituto Politécnico de Bragança, Campus de Santa Apolónia, 5300-253 Bragança, Portugal; 7Department of Chemistry and Biochemistry, School of Science and Technology (ECT), Universidade de Évora, 7000-671 Évora, Portugal; mel@uevora.pt

**Keywords:** nitrate, UV, optical, spectroscopy, real time, portable device, water, absorbance, opto-electronic, chemistry, environment, human health

## Abstract

Nitrates can cause severe ecological imbalances in aquatic ecosystems, with considerable consequences for human health. Therefore, monitoring this inorganic form of nitrogen is essential for any water quality management structure. This research was conducted to develop a novel Nitrate Portable Measurement System (NPMS) to monitor nitrate concentrations in water samples. NPMS is a reagent-free ultraviolet system developed using low-cost electronic components. Its operation principle is based on the Beer–Lambert law for measuring nitrate concentrations in water samples through light absorption in the spectral range of 295–315 nm. The system is equipped with a ready-to-use ultraviolet sensor, light emission diode (LED), op-amp, microcontroller, liquid crystal display, quartz cuvette, temperature sensor, and battery. All the components are assembled in a 3D-printed enclosure box, which allows a very compact self-contained equipment with high portability, enabling field and near-real-time measurements. The proposed methodology and the developed instrument were used to analyze multiple nitrate standard solutions. The performance was evaluated in comparison to the Nicolet Evolution 300, a classical UV–Vis spectrophotometer. The results demonstrate a strong correlation between the retrieved measurements by both instruments within the investigated spectral band and for concentrations above 5 mg NO_3_^−^/L.

## 1. Introduction

Monitoring aquatic systems is crucial for the management of water used for human consumption, aquaculture, recreational activities, irrigation, and industrial processes [1,2]. Nitrates (NO_3_^−^) are a common contaminant of surface waters [3,4,5]; therefore, they can pose severe risks to the environment [6,7] and human health [8,9,10]. Excessive amounts of NO_3_^−^ in water bodies may increase the risk of aquatic environment degradation such as eutrophication, resulting in the rapid growth of harmful algae blooms [11], cyanobacteria [12], and bottom anoxia [13], and can lead to an increase in atmospheric methane levels [14,15]. Drinking water contaminated by NO_3_^−^ is also correlated with fetal malformations during pregnancy [16,17] and the development of new-born methemoglobinemia [18,19] and may increase the risk of colorectal cancer due to the transformation of NO_3_^−^ into N-nitroso carcinogenic compounds [20,21]. The 91/676/EEC (Nitrate Directive), concerning the protection of waters against pollution caused by NO_3_^−^ from agricultural sources [22], transposed by Decree-Law 235/97 [23], defines an admissible concentration level of 50 mg NO_3_^−^/L in freshwater for human consumption. The target 6.3 water quality and wastewater from the United Nations Organization sets the objective for 2030 concerning the improvement of water quality to protect ecosystems and human health [24,25]. The same target refers to monitoring as a vital tool for policymakers and decision-makers to identify water bodies with high concentrations of pollutants. An affordable device for water NO_3_^−^ determination can also help developing countries as they can ensure the availability and sustainable management of water and sanitation for all [24].

There are several laboratory methods available that are used for the determination of NO_3_^−^ in drinking water, such as the spectrophotometry in the ultraviolet (UV) spectral region at 220 nm [26,27], the second derivative of the UV spectrum [28,29], ultraviolet screening [30], the NO_3_^−^ electrode [31], the cadmium [32] and hydrazine reduction [33], and the cadmium reduction flow injection [34]. Although these methods are precise and accurate, they have some disadvantages; for example, they require intensive time and laborious techniques [35], rely on expensive equipment [36], need to be operated in the laboratory, and involve the use of chemical reagents in colorimetric procedures. They also require the transportation of the samples to the laboratory, which can increase the risk of contamination by mineralization, nitrification/denitrification, fluctuations in temperature, and container handling [37,38]. In order to overcome these constraints, portable devices can be used directly in the field.

Portable spectral integrating devices can be used to quantify the absorbed radiation of the chemical compounds by varying the impedance or by converting the captured radiation into an electrical signal [39,40]. Several studies reported and quantified the presence of NO_3_^−^ in agricultural nutrient determination in wastewater and organic compounds using its typical absorption peak at 302nm as a proxy [41,42]. Traditionally, this peak is also used in the food industry for high concentrations, typically, above 1000 mg NO_3_^−^/L, because of the high linearity between the absorbance and the level of NO_3_^−^ present in the sample and the unnecessary use of reagents [43]. For lower concentrations, the n→π*weak absorption band of the NO_3_^−^ around 302 nm is far more challenging than using the traditional 220 nm NO_3_^−^ absorbance peak due to the lower absorbance, as reported in [44]. Some authors have used a band between 295 and 300 nm to correct the absorbance measured at 220 nm when a higher concentration of dissolved organic matter is present in the water sample [28]. In addition, this peak (302 nm) is not subjected to significant wavelength shifts (which can lead to increased error in the measurement of the NO_3_^−^ concentration) as the concentration increases [45,46]. Additionally, the UV LED introduced in this work presents a significantly lower commercial cost [47] compared to other UV LEDs near 220 nm on the market [48].

Moo et al. [47], as well as Szolga and Cilean [48], have applied a similar methodology for determining nitrate concentrations in water samples at a wavelength of 302 nm. However, the authors employed light sources that were not sufficiently optimized, and they used photodetectors with large detection windows without filters to eliminate unwanted interferences. Additionally, some have also used principal components analysis (PCA) to derive more than one parameter from the mixture without prior separation. They were unable to accurately determine the amount of NO_3_^−^ present in the samples for concentrations below 50 mg NO_3_^−^/L, as required by the European Nitrate Directive [22,49], the Nitrate Pollution Prevention Regulations Implementation of the Nitrates from the United Kingdom [50], and the National Primary Drinking Water Regulations from United States regulations [51]. Ingles et al. [52] developed a low-cost smartphone approach to determine the level of NO_3_^−^ in water samples. The instrument implies the usage of a scintillator to convert the UV light into visible green light and uses a setup based on a smartphone to record and process the signal. The instrument is more simple and compact and less expensive than the typical laboratory spectrometer.

This study presents the development and calibration of a new optical device called the Nitrate Portable Monitoring System (NPMS) based on low-cost optical and electronic components with highly accurate spectral characterization for NO_3_^−^ determination in water. This paper focuses on the initial validation of the NPMS instrument, which used laboratory-prepared NO_3_^−^ water samples at different concentrations. This procedure ensured controlled conditions to assess the baseline of the NPMS performance. It provided a test comparison performance benchmark to illustrate the accuracy, uncertainty, advantages, and limitations of the NPMS relative to existing technologies.

The uncertainties associated with the low-cost UV light-emitting diode (LED) were determined following the methodology outlined in Silva et al. [53] to assess the probability distribution, corresponding peak radiation wavelength, standard deviation, and suitability for use in NO_3_^−^ measurements. The principal specificity of the developed instrument is its ability to accurately detect and measure NO_3_^−^ concentrations in water samples while minimizing interference from other substances. This is achieved by using a paired diode–photodiode in the wavelength range that corresponds to the natural absorption peak of nitrate.

The design of the sensing system, its description in terms of hardware components, the characterization of the different optical and electronic parts, and the processing techniques are presented in Section 2. In Section 3, we describe the calibration of the developed NPMS and its performance in comparison to a benchtop laboratory spectrometer using a batch of NO_3_^−^ standard samples prepared in the laboratory. The best-fitting process and data analysis, as well as the evaluation of the uncertainties for both instruments, are presented. Section 4 discusses the results obtained, highlighting the unique features of the NPMS and the proposed future research.

## 2. Materials and Methods

### 2.1. Instrument Description

A conceptual diagram showcasing the paired diode–photodiode NPMS is presented in Figure 1. This diagram is divided into three subsystems: (i) the photon source module, which provides the radiation within the wavelength range of NO_3_^−^ spectral absorption; (ii) the sample support module, which accommodates the quartz cuvette containing the water sample to be analyzed and isolates the sample from the outside radiation to reduce possible interferents; and (iii) the processing and sensing module, where the spectra acquisition takes place, along with the implemented algorithm to determine the stability of the recorded signal and to retrieve the NO_3_^−^ concentration displayed on the integrated LCD. A computer can also be connected to the system to store the values in a permanent memory.

### 2.2. Photon Source

The instrument was developed considering the emission spectra of the LN-SMD3535UVB-P1 diode due to the characteristic emission radiation matching the absorption peak of NO_3_^−^ at 302 nm. The LED presents a spectral range between 295 and 315 nm with a peak at 308 nm. The diode is powered with a 7.5 V and interfaced with a 60 Ω load resistance, as shown in Figure 2, where the arrows indicate the radiation direction. To reduce the noise associated with the environment and to minimize the distance from the sensor, the diode was mounted at the entrance of the cuvette support.

Aiming to characterize the LED, its spectrum was acquired 5000 times using the SNT-BLK-CXR UV–Vis spectrometer through an optic fiber, as depicted in Figure 2. The measurements that were carried out allowed us to verify the device stability, statistical distribution, and parameter estimation. The chi-square test was applied to the recorded spectra to assess whether the radiation emitted by the LED follows a normal distribution. The spectral radiation intensity distribution showed a peak intensity at 308 nm within the range of 285 to 900 nm, as depicted in Figure 3.

The observed emission range from the LED was between 295 and 315 nm. Maximum and minimum intensities of 60,266 and 59,998 counts, respectively, were recorded. Spectral data, obtained from the output of the spectrometer analog-to-digital converter, are presented in digital counts. For the LED characterization, there is no need to convert them to radiation unity. The spectrometer was operated with an integration time of 57 ms.

The frequency distribution of the LED emitted intensity (blue bars) and the respective Gaussian distribution function (red curve) are presented in Figure 4. The chi-square test determined that the upcoming radiation from the LED follows a normal distribution for a *p*-value of 0.5. The reading of the average intensity was 60,112 counts, with a standard deviation of ±40 counts. Several tests were carried out to verify the reliability of the measurements.

### 2.3. Processing and Sensing System Design

The developed standalone device for detecting NO_3_^−^ is based on a sensing system composed primarily of the GUVA-S12SD UV sensor [54]. The primary function of the probe is to convert the incoming radiation emitted by the diode into an electrical signal to be processed by the microcontroller. The sensor is a low-cost UV Schottky-type photodiode with gallium nitride-based material, and it is usually applied for the detection stage [48]. The spectral response covers the UVA range and part of the UVB range from 240 to 370 nm. The main features of the photodiode are reported in Table 1.

Due to the low light intensity produced by the LN-SMD3535UVB-P1 LED and low absorbance of the NO_3_^−^ at 302 nm, the built-in sensor op-amp SGM8521 [55] was combined with the LM358 op-amp in a two-stage configuration. This amplifier can be supplied with a voltage ranging from 3.5 to 24 V, and it has a maximum electric DC output of 20 mA and a gain of 1000 with the two stages combined. The schematic illustration for the single beam UV radiation, the quartz cell for water sampling, and the sensing module coupled with the op-amp is presented in Figure 5.

The spectral response of the sensor was reduced to the range of the diode, highlighted with the blue dashed line (Figure 6a), isolating the sensor from the outside environment to match the diode spectral response (red curve) and the NO_3_^−^ peak. Figure 6b shows the linear response of the sensor for UVA optical radiation.

A Raspberry Pi microcontroller (model RP2040 [56]) was adopted for data acquisition and digital processing. The microcontroller architecture is based on dual Cortex M0+ processor cores, up to 133 MHz, 264kB of embedded SRAM in 6 banks, 30 multifunction GPIO, 2 SPI controllers, 4 multiplex channel 12-bit ADC with a sampling frequency of 500 kHz, and a 1.1 USB host/device. The MicroPython Pycharm interface was adopted for signal acquisition and data processing.

### 2.4. Enclosure Box and Printed Circuit Board

The electronic components described above are positioned in the specially developed PCB, which is contained in a box designed in SolidWorks and 3D-printed using black polylactic acid (PLA) filament. The black color of the filament intends to maximize the absorption of the scattered radiation inside the box. The various parts of the NPMS case are highlighted in Figure 7a, namely, the upper cover (1), the cap to cover the cuvette on the cell support (2), the cell support (3), the battery housing (4), the front cover (5), the PCB and electronic components seat in (6), and the LED placed at 1.75 cm from the bottom of the cuvette in the cut-out horizontally aligned with the photodiode (7).

All the electronic components of the device were assembled on a simple 2-sided PCB, as shown in Figure 8a. The LCD screen was plugged into the screen connector (1), the Raspberry Pi Pico was surface-mounted on the pad (2), the two-stage connector was used for the op-amp (3), the UV sensor was connected to (4), and the temperature and relative humidity sensor was connected to the pad (5). Figure 8b shows a photo of the physical PCB, produced using FR-4 TG 130–140 material, with a thickness of 1.6 mm and a black solder mask. 

The assembled enclosure box in different views is presented in Figure 9. The top view of the instrument operating in standalone mode is shown in Figure 9a. The different components can be easily identified: the LCD Module (1), the Raspberry Pi microcontroller (2), the LM358 Two-stage op-amp (3), the GUVA-S12SD UV sensor (4), the AHT10 humidity, the temperature sensor (5), the quartz cell for water samples (6), and the LED housed over the front cover (7). Figure 9b presents the front view of the instrument, displaying the power source connectors for charging and powering purposes. A general upper view of the instrument is shown in Figure 9c. Finally, the lateral view (Figure 9d) allows the identification of the engraved entry in the center of the USB connection (1).

### 2.5. Samples: Nitrate Standard Solution Preparation

The standard solutions for the calibration samples were prepared with ultrapure water Milli-Q and NaNO_3_^−^ Alfa Aesar (99.0% minimum purity crystalline). Previously, decontaminated labware was used. Six standard solutions of 5, 10, 25, 50, 75, and 100 mg NO_3_^−^/L were prepared to cover a wide range of concentrations below and above the regulation limits. The mass of NaNO_3_^−^ for the solutions was obtained from stoichiometry calculations.

From the stock solution, the adequate volumes were extracted and diluted to 100 mL with ultrapure water to prepare the standard solutions of interest. From the stock solution, 10, 7.5, 5, 2.5, 1, and 0.5 mL amounts were extracted and diluted to 100 mL with ultrapure water to prepare the standard solutions of interest. To prevent the degradation of the NO_3_^−^ samples throughout the experiment, all the solutions were stored below 2 °C.

### 2.6. Calibration Algorithm

The methodology for the NPMS calibration foresees the double measurements of the blank sample containing only ultrapure water ($I_{0}$) with each one of the samples with different NO_3_^−^ concentrations ($I_{k}$). The sensor output is measured 24.000 times, aiming to reduce the signal-to-noise ratio (SNR) and, therefore, to obtain results with better accuracy. For each acquired dataset, the voltage value produced by the UV sensor is proportional to the intensity of the radiation that passes through the water sample. The ADC of the microcontroller reads this voltage, and the signals are filtered using the following algorithm: a chi-square test is applied to prevent the acquisition of random or aberrant signals. In this step, if the null hypothesis $h_{0}$ is rejected, the collected measurement is also rejected. A simple running average is applied to smooth the acquired signal, as shown in Equation (1).
(1)$$\left\{\begin{array}{c} s_{1}=\frac{1}{n}\sum_{j=1}^{n}a_{j}\\ s_{i}=\frac{1}{2n}\sum_{j=n\left(i-2\right)+1}^{i\times n}a_{j} \end{array}\right.$$

When n is equal to 6000, i varies between $\left\{2,\ldots ,4\right\}$, $a_{j}$ is the measured data set inside each section and $S_{i}$ is the moving average of each section. Then, the initial intensity $I_{0}$ and the sample intensity $I_{k}$ are calculated as the average of ${\left\{S_{i}\right\}}_{i=1}^{i=4}$. The fragmentation of the dataset in multiple intervals (4) is due to hardware restrictions since this method allows the bypassing of the memory ram limitation of the RP2040 microcontroller and increases the number of measurements available for the noise filtering process.

Then, the absorbance is calculated using the following expression:(2)$$I_{k}\left(\lambda \right)=I_{0}\left(\lambda \right)\exp\left(-\sigma \left(\lambda \right)cL\right)\Rightarrow A=-\log\left(\frac{I_{k}\left(\lambda \right)}{I_{0}\left(\lambda \right)}\right)\;k=1,2,\ldots ,6$$
where $I_{k}$ denotes the radiation intensity of the beam after passing through the water sample, $I_{0}$ is the initial intensity of the blank sample, which includes the cell and ultrapure water, A is the absorbance, σ is the molar absorptivity, λ is the wavelength, c is the molar concentration of the substance, and L is the length of the light path. The UV sensor produces an analog value (in mV) that is converted into digital counts from the AD converter on the microcontroller. Since this is integrated over the spectral range considered (295–315 nm), the digital signal (DS) obtained from the AD is converted in voltage (Vs) using the following equation: (3)$$V=DS*3.3/2^{ADC_{BN}}$$
where 3.3 is the reference voltage and ${ADC}_{BN}$ is the number of bits of the ADC (in our case, ${ADC}_{BN}=12$).

Using voltage levels recorded from the UV sensor, the absorbance A is as follows:(4)$$A=-\log\left(\frac{I_{k}\left(\lambda \right)}{I_{0}\left(\lambda \right)}\right)=\log\left(\frac{V_{BS}}{V_{NS}}\right)$$
where $V_{BS}$ is the measured voltage of the blank sample using ultrapure water and $V_{NS}$ is the measured voltage of the water samples containing NO_3_^−^. To obtain the result of the absorbance per sample, a simple average is applied to the acquired signal. Then, the absorbance results are stored in a table file. The flowchart in Figure 10 presents the developed methodology for the calibration of the NPMS.

In order to produce the calibration equation for the determination of NO_3_^−^, the absorbance data are correlated to the known concentration of the standard solutions following the Beer–Lambert law [57], according to the following equation:(5)$$A=-\sigma \left(\lambda \right)\;c\;L$$

The best-fit function for the measured data is obtained with linear regression, as presented in the equation below:(6)Y=βX+ε
where *Y* represents the vector of the measured concentration, X is the matrix of the measured absorbance, ε is the error of the estimative, and β is the vector of the parameters of the linear model to be estimated, which is expressed in Equations (7) and (8).
(7)$$X=\left(\begin{array}{cc} 1 & A_{1}\\ \vdots & \vdots \\ 1 & n \end{array}\right)\;\mathrm{with}\;n=6;$$
(8)$$\beta =\left(\begin{array}{c} \beta_{0}\\ \beta_{1} \end{array}\right)$$

The linear regression model was estimated using the ordinary least squares method. This approach relies on minimizing the sum of the squares residuals and allows the estimation of the vector of parameters β. The solution β for the linear least squares fit is as follows:(9)$$\beta ={\left[X^{T}X\right]}^{-1}X^{T}Y$$

### 2.7. Processing Algorithm for Standalone Measurements

The determination of NO_3_^−^ in water samples is performed using the following procedure. First, a blank sample comprising ultrapure water is placed in the cuvette support to establish the necessary instrument offset. Subsequently, the algorithm follows the same procedure adopted for the NPMS calibration. The absorbance of the sample is determined using Equation (4), and the NO_3_^−^ concentration in the individual sample is calculated via the interpolation of the linear equation derived during the calibration process. The result is then displayed on the integrated LCD screen. Consequently, it is not necessary to connect a PC to the instrument, facilitating field measurements. The flowchart outlining this procedure, including the algorithm, is depicted in Figure 11.

### 2.8. Chemical Reagents and Fe^2+^ Samples

To evaluate the performance of the NPMS, the results of the measured absorbance were compared to the benchtop laboratory spectrophotometer Nicolet Evolution 300 by Thermo ELECTRON CORPORATION UV–Vis, controlled by VISIONpro PC Control Software (Vision Version 4.10). The Nicolet Evolution 300 has two quartz cells with a light path length of 10 mm and a height of 300 mm. One cell is loaded with the blank sample, and the second is loaded with the standard solutions at concentrations equal to those of NO_3_^−^. Table 2 presents the characteristics of the spectrophotometer.

To determine the NPMS performance, the standard curve, R-square coefficient, Pearson correlation, and F-test were assessed. In the experiment, we first measured the spectra and determined the absorbance for NO_3_^−^ samples at 302 nm using the reference instrument and then using the NPMS system. The flowchart shown in Figure 12 was implemented to determine the absorbance of NO_3_^−^ in the prepared samples. In order to maintain the stability of the measurements, the cell was kept in the support, and the water samples were exchanged using a syringe.

## 3. Results and Discussion

### 3.1. Sensitivity Analysis

The linear regression properties, including the coefficient of determination, the coefficients of the linear least squares equation, and the root mean square error (RMSE), are presented in Figure 13. The relation between the variances of the two instruments was determined using the F-test. An uncertainty analysis with a confidence interval of 96% was performed to determine the instrument’s accuracy.

The linear function had a slope of 1.121 and an intercept of 4.02 × 10^−4^, indicating a strong relation between both devices regarding the measured absorbances. This is also shown by the determination of R^2^, with a value of 0.975. Overall, an absorbance overestimation of the NPMS is observed compared to the Nicolet Evolution 300 spectrophotometer, with a value of 2.45%. The RMSE reached a magnitude of 1.24 × 10^−3^, and the Pearson coefficient was 0.987, indicating a strong positive linear relationship. The result of the F-test presents an equality in the variance of both instruments for a 5% significance level. Table 3 presents the absorbance measurements for the NO_3_^−^ standards as well as the F-test, variance, and critical values.

### 3.2. Results for the Calibration of the NPMS

The NPMS was calibrated using a series of standard NO_3_^−^ solutions ranging from 5 mg NO_3_^−^/L to 100 mg NO_3_^−^/L. Once again, a precise linear relationship between the standard solutions and the measured absorbance was found. The scatter plots in Figure 14 show the absorbance of the NO_3_^−^ measured with the NPMS compared to the standard solutions; it also shows the calibration line obtained from the experimental data using linear least squares fit, which also allows the retrieval of the unknown concentrations from the measured absorption. Furthermore, the R^2^ coefficient and the value of the RMSE are reported to assess the fitting quality.

The abscissa represents the recorded absorbance, and the ordinate axis represents the concentration of the standard samples prepared in the laboratory. The linear equation with a slope of 8350 and intercept of 1.797 was obtained and was the relationship used by the instrument to retrieve the NO_3_^−^ concentration. The R2 value of 0.988 shows a good relation between the prepared NO_3_^−^ samples and the absorbance, with a calculated RMSE of 3.789.

The calibration curve for the classical Nicolet Evolution 300 spectrophotometer at a wavelength of 302 nm is presented in Figure 15. The linear equation with a slope of 7317 and an intercept of 4.06 was used to retrieve the NO_3_^−^ concentration from the recorded absorption. The instrument also shows an R^2^ of 0.979 and an RMSE of 5.15. Comparing the R^2^ and the RMSE of the two instruments, it can be observed that the low-cost NPMS has a better performance in retrieving the NO_3_^−^ at wavelengths ranging between 295 and 315 nm than the Nicolet spectrophotometer due to a better RMSE and standard deviation.

## 4. Conclusions

The development of a low-cost specialized NO_3_^−^ portable measurement system (NPMS) has been successfully achieved for water samples. We have demonstrated the validity of using the 302 nm absorption peak for NO_3_^−^ determination in aqueous solutions. The use of a UV diode emitting radiation between 285 and 315 nm, paired with a photodiode sensing from 240 to 360 nm, allows the measurement of NO_3_^−^ absorption on the optimal spectrum range without the interference of other chemical compounds or stray light. Thanks to this option, the cost of the device is significantly lower than it would be if the literature-recommended wavelength of 220 nm were used due to the higher cost associated with the typical source for this spectral range.

The characterization of the LED performance shows a stable photon source characterized by a normal distribution and a standard deviation of ±40 counts. Although the standard errors of the measurements might initially seem high, the developed system presents smaller standard errors than the reference benchtop spectrophotometer. Several methods like shielding insulation, signal averaging, and a low-pass filter could be introduced to improve the accuracy of the measurements.

The developed case allowed for the accommodation of all the instrument components in a small and robust footprint device that can be transported and deployed in the field for real-time measurements. A comparison study of the NPMS and a classical Nicolet spectrophotometer was developed, and the results showed a good agreement between the two devices, with a high R^2^ of 0.975. The F-test also presented a high correlation for the variances of the six NO_3_^−^ standard samples, as did the Pearson correlation test, with a value of 0.987. Further work will be conducted to study the influence of the temperature on the system and to integrate a photon source with different wavelengths to allow multi-component detection and cross-interference corrections in particulate organic matter at a wavelength of 275 nm and to apply the developed instrument to determine NO_3_^−^ concentrations in freshwater samples.

## Figures and Tables

**Figure 1 sensors-24-05367-f001:**
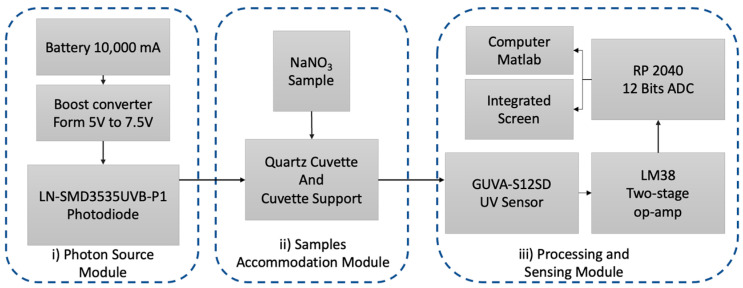
Block diagram of the nitrate measurement system, focusing on the main part of the developed NPMS and its interactions.

**Figure 2 sensors-24-05367-f002:**
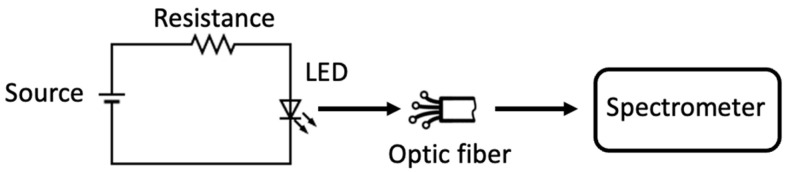
Light emission diode measurement scheme.

**Figure 3 sensors-24-05367-f003:**
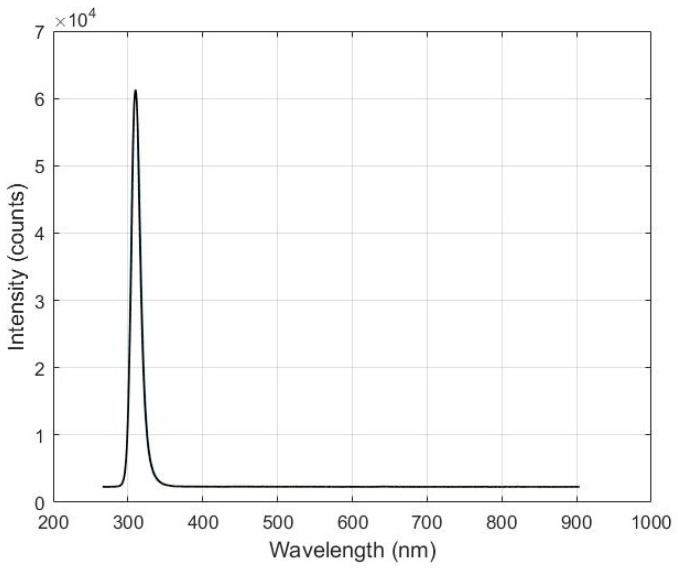
Spectral energy distribution of the LN-SMD3535UVB-P1 LED with peak emission at 308 nm.

**Figure 4 sensors-24-05367-f004:**
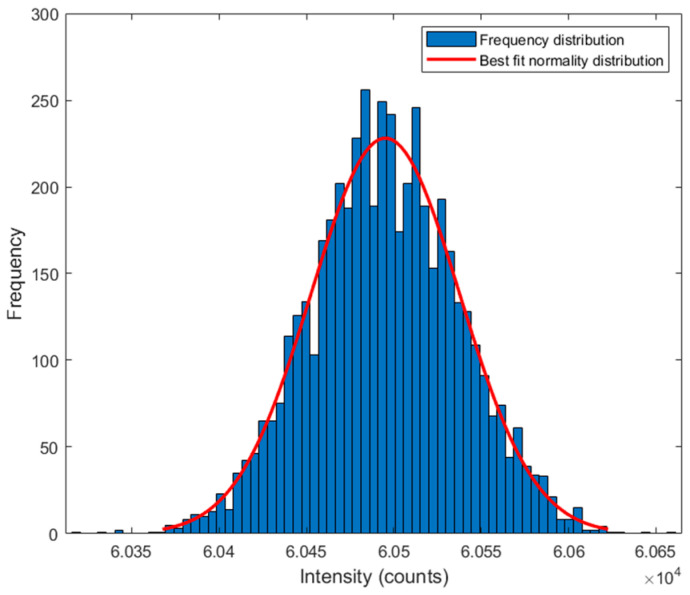
Histogram of the normal distribution of the 308 nm peak for 1000 measurements.

**Figure 5 sensors-24-05367-f005:**
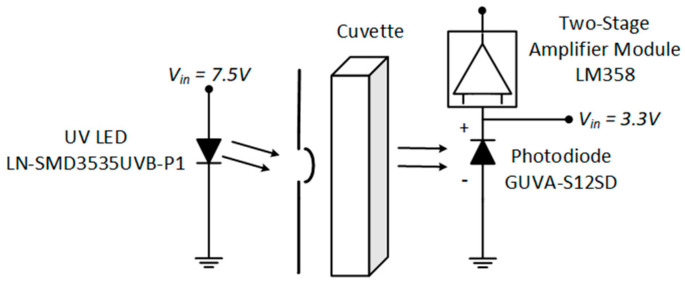
Schematic diagram of the instrument for emission radiation, the quartz cell, the photodiode, and the two-stage amplifier module.

**Figure 6 sensors-24-05367-f006:**
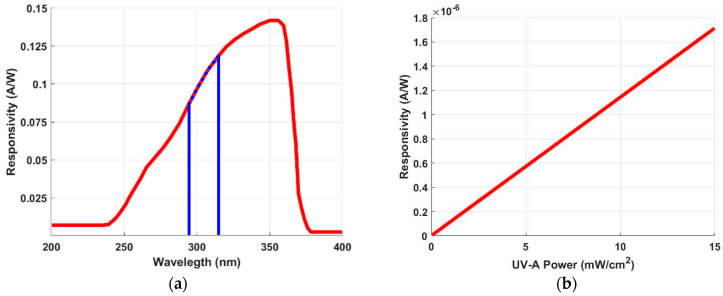
(**a**) UV sensor response at intensity radiation between 240 and 370 nm (red line) and the diode wavelength range emission (dashed blue line). (**b**) Sensor photocurrent response when irradiated with UVA radiation [54].

**Figure 7 sensors-24-05367-f007:**
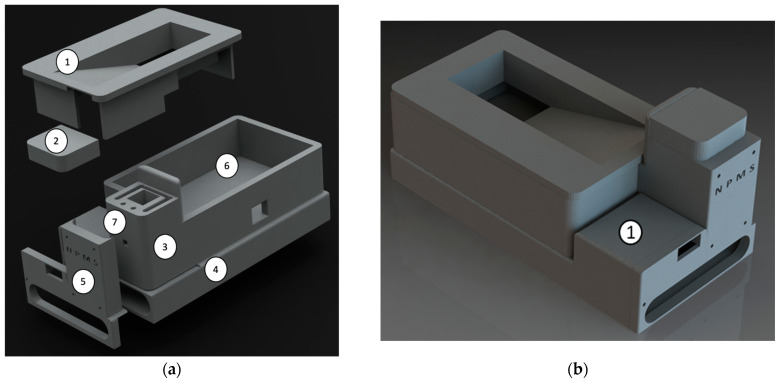
(**a**) Exploded view of the 3D model case developed in SolidWorks that accommodates all the NPMS components. (**b**) Rendering of the NPMS case assembled.

**Figure 8 sensors-24-05367-f008:**
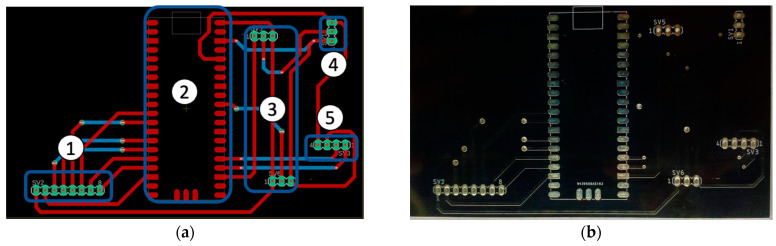
(**a**) Diagram of the top side of the printed circuit board, showing the component orientations. In this diagram, the blue lines indicate the vias placed on the bottom surface of the board, while the red lines indicate the vias placed on the top surface. (**b**) PCB ready for the assembly of the electronic components.

**Figure 9 sensors-24-05367-f009:**
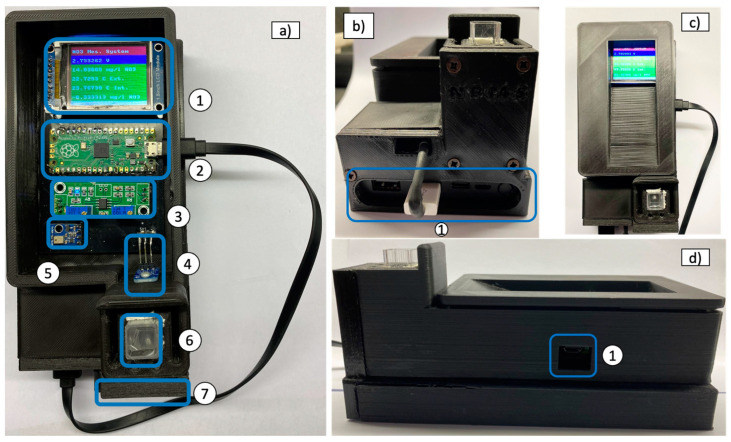
(**a**) Top view of the device NPMS configured for standalone use. (**b**) Front view of the instrument. (**c**) Top view of the device fully assembled. (**d**) Lateral view of the instrument.

**Figure 10 sensors-24-05367-f010:**
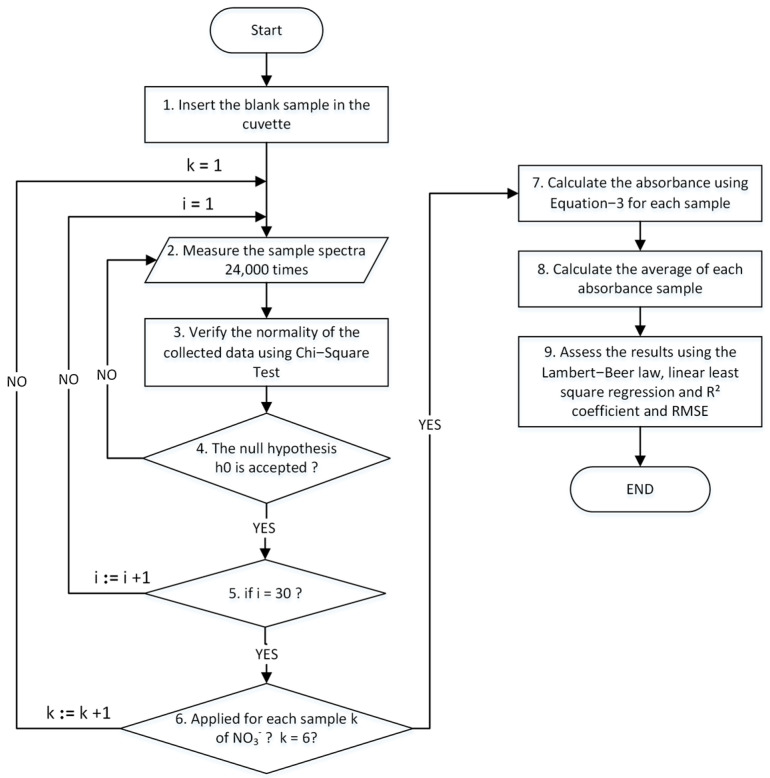
Flowchart for the measurement methodology using the NPMS.

**Figure 11 sensors-24-05367-f011:**
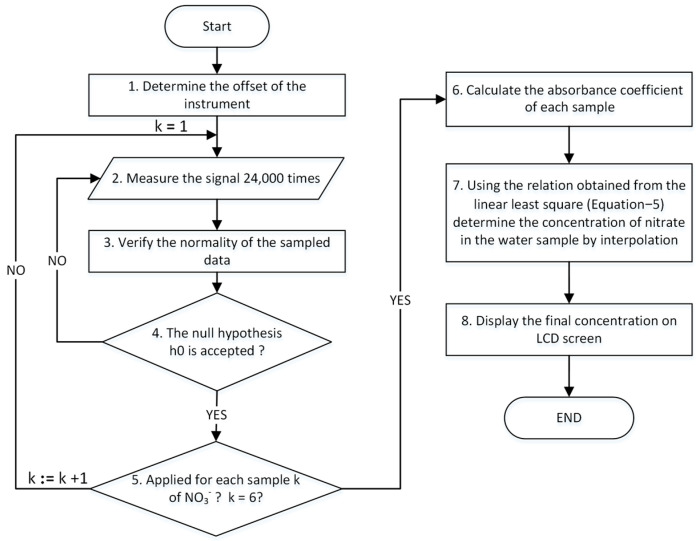
Flowchart of the methodology implemented by the NPMS.

**Figure 12 sensors-24-05367-f012:**
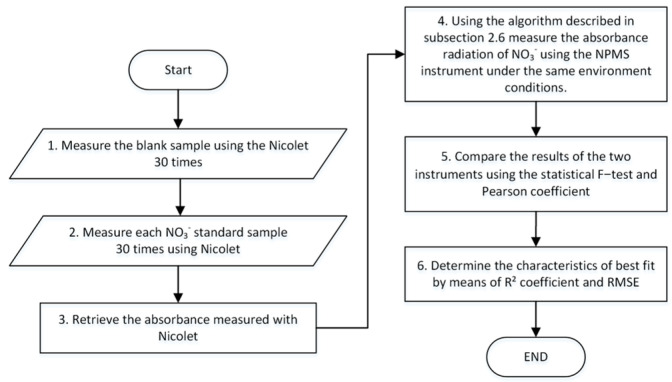
Flowchart of the implemented methodology to compare both instruments.

**Figure 13 sensors-24-05367-f013:**
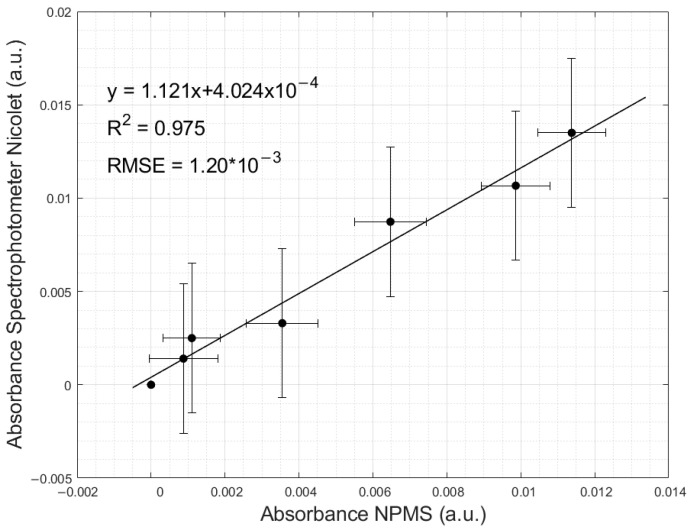
Absorbance retrieved with the NPMS towards the absorbance measured with the Nicolet spectrophotometer.

**Figure 14 sensors-24-05367-f014:**
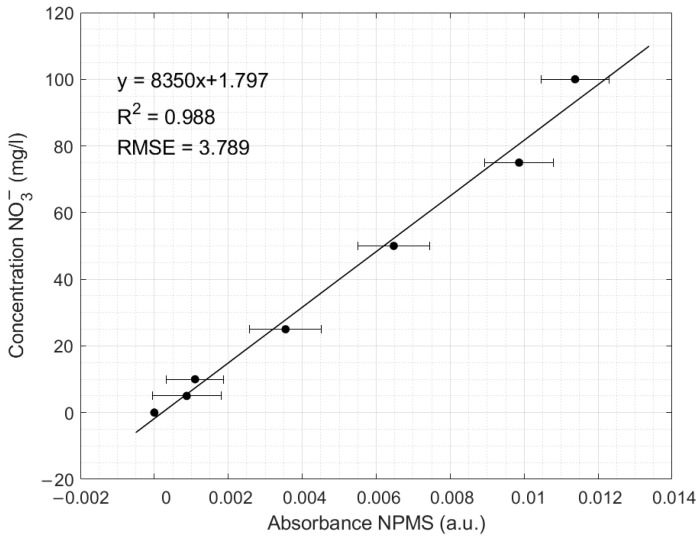
NPMS transfer function showing the integration of the recorded absorbance spectra of NO_3_^−^ using the NPMS.

**Figure 15 sensors-24-05367-f015:**
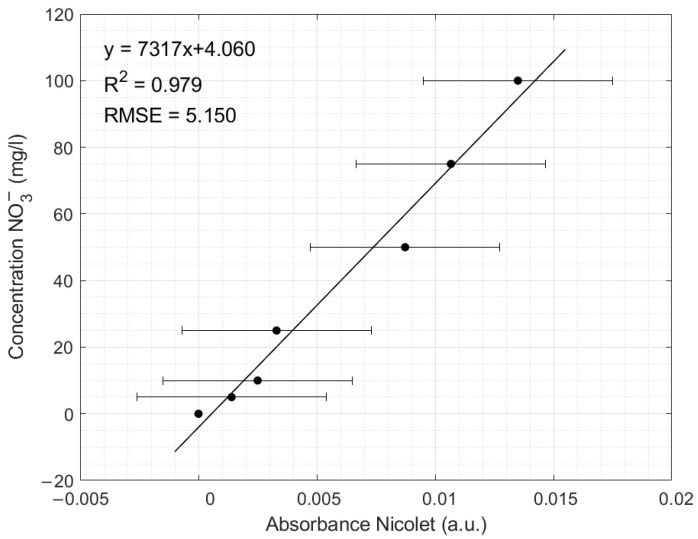
The transfer function of the classical Nicolet spectrophotometer at wavelength 302 nm.

**Table 1 sensors-24-05367-t001:** Photodiode electrical characteristics.

Parameter	Value
Forward current	1 mA
Reverse voltage	5 V
Working voltage	2.7 V to 5.5 V
Active area	0.076 mm^2^
Typical dark current at 25 °C with VR of 0.1 V	1 nA
Photocurrent with UVA Lamp of 1 mW/cm^2^	113 nA
Temperature coefficient	0.08%/°C
Responsivity at λ = 300 nm with VR of 0 V	0.14 A/W
Operation temperature	−30 °C to 85 °C
Spectral detection range	240 to 370 nm

**Table 2 sensors-24-05367-t002:** Nicolet Evolution 300 spectrophotometer characteristics.

Parameter	Value/Unity
Holographic grating	1200 lines/mm, blazed at 240 nm
Maximum resolution	0.5 nm
Range	190 to 1100 nm
Accuracy	±0.20 nm (546.11 nm Hg emission line) ±30 nm (190 to 900 nm)
Repeatability peak separation of repetitive scanning of Hg line source	<0.10 nm
Standard deviation of 10 measurements	<0.05 nm
Accuracy of instrument	1A: ±0.004 A 2A: ±0.004 A 3A: ±0.006 A
Repeatability of light intensity measurement	1A: ±0.0025 A
Drift	<0.0005 Abs/hour at 500 nm, 2.0 nm SBW, 2 h warm-up
Baseline flatness	±0.0015 A (200–800 nm), 2.0 nm SBW, smoothed

**Table 3 sensors-24-05367-t003:** Results for the measurement standards obtained with the NPMS and Nicolet Evolution 300.

Sample (mg NO_3_^−^/L)	Mean Abs. NPMS	Mean Abs. Nicolet	Variance NPMS (mg NO_3_^−^/L)	Variance Nicolet (mg NO_3_^−^/L)	Critical Value
5	0.0008	0.0014	2.1 × 10^−7^	2.4 × 10^−7^	1.10
10	0.0011	0.0025	1.5 × 10^−7^	2.5 × 10^−7^	1.65
25	0.0035	0.0033	2.3 × 10^−7^	2.1 × 10^−7^	0.89
50	0.0064	0.0087	2.3 × 10^−7^	1.9 × 10^−7^	0.82
75	0.0098	0.0106	2.1 × 10^−7^	2.2 × 10^−7^	1.01
100	0.0113	0.0135	2.0 × 10^−7^	2.5 × 10^−7^	1.19

## Data Availability

Data are contained within the article.

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
