# Peer review of "The Development of a Novel Nitrate Portable Measurement System Based on a UV Paired Diode–Photodiode"

_sensors, 2024, doi:10.3390/s24165367_

Round 1

Reviewer 1 Report

Comments and Suggestions for Authors

This work describes the development and testing of a low-cost system for detection of NO3- in water samples, and its comparison with a standard spectrophotometer system.

In general, the methodology and experimental steps are well and clearly described.  The experimental data is also well documented and presented.

The manuscript still requires a revision of format to be in accordance with the journal standards.  For example, the way how references are given in the text appeared in some cases wrong (e.g., line 62). In other cases, there are spaces missing (e.g., line 63).  The format used in the References section is also not equal in all references presented.  These are just some punctual examples.  I suggest a general review of the final format style of the manuscript.

Despite of the well-presented experimental data, in my opinion there is a crucial point still missing in this work.  The authors claim that the main advantage of their new NPMS system is based on its portability and flexibility for in-field NO3- detection.  It is also commented on the advantages of their system in comparison with other several laboratory methods.  one of them been precisely the possibility of measurement of 'real' samples on the field without the need for transporting samples to the laboratory.  By at the end, the measurements presented here to validate the system are standard samples prepared and measured in laboratory.  It is clear that a proper calibration must be done, and also important, a comparison with another standard test system.  But this is, in my opinion, half of the story.  The other missing part of the story should present the performance of the NPMS in field situation and real samples.  Otherwise, the whole point of a novel and portable system is missed.  In not doing in-field investigations, other very important aspects of the characterization are missing.  For example, temperature influence.  Evidently, temperature will affect the performance of the system for NO3- detection, especially because all of the optical components involved here will also be affected by the temperature effects.  Therefore, I believe that temperature investigations should not be mentioned as further work in the Conclusions section.  Instead, when presenting a novel system which is expected to be an alternative to already established detection systems, temperature investigations and real samples measurements should be part of the basic system characterization.  The merit of the novelty should not be based only on the low-cost advantages.  Of course, such advantages are very important.  But, this is not of any benefit if the system is not proven to perform properly in real field conditions.  I encourage the authors to first consider including such investigations for a proper validation of the system.

Author Response

Thank you for the detailed and constructive comments to the manuscript above referred and for the encouragement for the inclusion of in-field measurements for a proper validation of the system. We appreciate the reviewer recognition of the well-presented experimental results, as well as the need for validation of the NPMS system. We acknowledge that our current manuscript is primarily focused on laboratory-prepared standard samples and also understand the importance of validating the system under real field conditions to better demonstrate its portability, flexibility and usefulness. The reviewer outstands four main topics that will be discussed further: In-situ performance and determination of NO3- in real samples, Temperature Influence Over Measurements, Comparison of the NPMS with Standard Test Systems and Future Work.

In-Situ Performance and Determination of NO3- In Real Samples

We agree with the reviewer that a new system performance should be tested in the real environment to validate its practical applicability. In our ongoing work, we have planned a series of tests to assess the NPMS system under various environmental conditions extensively. These measurements will provide a better understanding of the system robustness and reliability outside the laboratory. The determination of NO3- under laboratory ideal conditions allowed us to demonstrate that with the developed low-cost instrument, it is possible to determine nitrates concentrations spectral range between 295 and 315 nm with high accuracy and precision up to a concentration as low as 5 mg NO3-/l. As for all instruments, the application to real environments can introduce a plenitude of challenges to solve that can be to other interference compounds or higher opacity of the sample. If there is a suspicion of interference compounds, they should be removed from the sample as described in [1]. For samples with higher opacity, as the absorbance will tend to be higher, some methods such as filtration, dilution in series and centrifugation, should be used to eliminate.

Temperature Influence Over Measurements:

The reviewer correctly addresses the need to study the influence of temperature on the instrument as it can significantly impact the performance of optical and electronic components in our detection system. These experiments have the goal of characterizing the temperature dependence on the instrument and the development of calibration strategies such as temperature compensation algorithms, temperature-stabilized components, environment enclosure, calibration curves and numerical models.

Due to the system novelty and the topic relevance focused on the development of a low-cost paired-diode-photodiode that can determine NO3- in water samples on a very challenging radiation band and without the need for reagents, the In-Situ Performance and Determination of NO3- In Real Samples and Temperature Influence Over Measurements, studies must be carried out extensively. These findings should be detailed comprehensively in two dedicated manuscripts rather than introduced here almost as a minor study. The present manuscript main goal was to address the development of the NPMS.

Comparison of the NPMS with Standard Test Systems

The original manuscript focuses on the initial validation of the NPMS instrument, which used laboratory-prepared NO3- water samples at different concentrations. This procedure ensured controlled conditions to assess the baseline of the NPMS performance. We have provided a test comparison performance benchmark in order to illustrate the accuracy and uncertainty and to illustrate the advantages and limitations of the NPMS relative to existing technologies.

[1]         E.W. Rice, R.B. Baird, A.D. Eaton, L.S. Clesceri, Standard methods for the examination of water and wastewater, American public health association Washington, DC, 2012.

As the reviewer requested, we have now further changed the format of the references in lines 62 and 63. A revision of the manuscript has been, and we believe it now aligns with the requirements of the reviewer.

Reviewer 2 Report

Comments and Suggestions for Authors

The manuscript represents a low-cost and reagent-free portable nitrate measurement system. The system is very sample-sampling and the results show that its ability to detect nitrate is similar to that of Nicolet Evolution 300. I think the manuscript can be accepted for publication in Sensors. I have the following questions.

1. Low cost is an important feature of NPMS. What is the cost of NPMS and how does the cost of NPMS compare to other reports or commercial systems.

2. In the manuscript, NaNO3 is used to prepare the solution. In natural water, there are many other components such as organic matter, salts, and solid particles. I wonder if the absorbance will change if there are other components in the solution?

Author Response

We want to thank the reviewer for your revision and encouraging comments. Below, we address the key points raised:

  1. The NPMS cost estimate is 65€. It can be considered an ultra-low-cost instrument when compared with benchtop spectrometers. For instance, the device used in comparison with the NPMS cost around 25.000€.

  1. As the reviewer has noticed, in our experiments, we used NaNO3 in order to prepare the water solution. The main goal of the manuscript was the development of a low-cost instrument based on UV radiation for NO3- determination. It is indeed true that natural water sources contain a multiplicity of contaminants/interferents, such as salts, organic matter, solid particles, and microscopic living organisms. These components can lead to deviations in absorbance from the real concentration of NO3-. The determination of chemical compounds in water is subject to this concern, as reported in many methods presented in [1]. In such cases, when the user doubts the existence of interferents, removing them from the final reported concentration is necessary. This can be carried out using two main methods: physically removing the interferents from the sample and numerical by subtraction of the total interference from the total measurement:

1: Physical. Using reduction columns, filtration, centrifugation, coagulation and flocculation, sedimentation, adsorption, precipitation, series of dilutions, matrix matching, capillary electrophoresis and standard addition. After removing all the interferents from the sample, the NO3- concentration should be correctly determined. Typically, organic matter impacts the measured absorbance along all the UV/Vis spectra. Due to that high concentrations of organic should be removed from the water sample before measurement of NO3-. However other chemical compounds that do not absorb in the spectra range from 295 to 315 nm will not cause a bias on the NO3- determination.

2: Numerical. The second procedure to correctly determine the concentration of NO3 in the presence of interferents is to determine the concentration of substances producing a bias using alternative methods. This can include determining the interferents using, for example, conductometry, potentiometry, coulometry, trituration, and ion chromatography. After the concentration of the interferents is determined, it should be subtracted from the final concentration, and the amount of NO3- can be obtained.

Investigators and analysts regularly use these methods to determine substance concentrations, including when working with advanced inductively coupled plasma devices.

[1]         E.W. Rice, R.B. Baird, A.D. Eaton, L.S. Clesceri, Standard methods for the examination of water and wastewater, American public health association Washington, DC, 2012.

Reviewer 3 Report

Comments and Suggestions for Authors

Nitrates are contaminants which may cause severe risks. This manuscript describes a portable measurement system for nitrate using UV paired diode-photodiode. This work shows the practical detection of nitrates, which demonstrates the potential for commercial applications. Below are the comments:

1. The manuscript describes the system can detection concentrations above 5 mg NO3-/l. What are the minimum nitrate concentrations allowed in different samples in relative regulations or rules? Does the detection capacity of the system meet the requirement?

2. What is the difference between this design and the strategies in other studies? please highlight in the main text.

3. What is the specificity of the system? Please demonstrate. 

4. The range of detection is 5 mg NO3-/l-100 mg NO3-/l, which seems narrow. Can the system detection a wider range of analytes?

5. The standard error values seem high. Is it any way to reduce the values? Please comment.

Comments on the Quality of English Language

The English Language is general good.

Author Response

Thank you very much for your revision and encouraging comments.

  1. According to the Council Directive of 12 December 1991 concerning the protection of waters against pollution caused by nitrates from agricultural sources (91/676/EEC), the limit of nitrate in drink water should be no more than 50 mg NO3-/l. The instrument presents a Quantification Limit of 5 mg NO3-/l, which is 10 times lower than the limit in the Directive.

  1. The main difference between this design and the other studies is that there is no need for chemical reagents to determine NO3- concentrations in water as it used the natural n→π*weak absorption band of the NO3- around 302 nm. As demonstrated in the manuscript, the instrument also presents high accuracy and precision in the determination of NO3-, performing as well as or better than a more expensive benchtop spectrometer.

  1. The main specificity of the developed instrument is the ability to accurately detect and measure NO3- concentrations in water samples while minimising the interference from other substances. This is achieved by using a paired-diode-photodiode in the range of wavelengths that correspond to the natural absorption peak of nitrate. The NPMS was compared and calibrated against the bench-top spectrophotometer Nicolet Evolution 300 UV-Vis. The comparison shows a high correlation coefficient (R² = 0.975), which is an indicator that the NPMS accurately determines nitrate with similar performance as the laboratory-grade spectrophotometer. The device presents a small form factor which provides an easy way to transport it to the field. Additionally, is a self-contained instrument, all the components are integrated.

  1. The range detection was designed with the main goal of providing an instrument capable of determining the NO3- in accordance with the Nitrate Directive (91/676/EEC), which imposes a limit of 50 mg NO3-/l for drinking water. The absorbance for the 100 mg NO3-/l is 0.0119.

The maximum absorbance that an optoelectronic instrument can determine without bias is 1. The Beer-Lambert law defines the absorbance between 0 and 1 to determine the concentration of a chemical substance. In this range, the measured absorbance varies linearly. The variation is exponential for absorbances higher than 1 and should not be used. Following the theory and admitting a linear variation for an absorbance of 1, our system could determine around 8403 mg NO3-/l.

The system was developed initially for the determination of NO3-, in the range between 295 and 315 nm. Perhaps if there is a substance that presents an absorbance in the wavelength range, the system can be calibrated for it.

  1. The standard errors could seem high for the measurements. However, the developed system presents standard errors smaller than the reference bench-top to the spectrophotometer. There are several methods that can be implemented in order to improve the accuracy of the measurements:

  1. Shield insolation—introducing an electromagnetic shield could decrease the noise interference from the surrounding electronic components and consequently improve the instrument's accuracy.

  1. Signal Averaging—increasing the average of measurements per water sample to reduce the signal-to-noise ratio (SNR) and smooth fluctuations could decrease the standard errors.

  1. Filtering— implementing a low-pass filter can decrease the high-frequency noise without significantly affecting the signal quality. This would lead to better uncertainty in the measurement and lower standard error.

Round 2

Reviewer 1 Report

Comments and Suggestions for Authors

No extra comment from my side.

Reviewer 2 Report

Comments and Suggestions for Authors

Thanks for the authors' responses, I have no more questions.